# Development of oats flour and bitter gourd fortified cookies: Effects on physicochemical, antioxidant, antimicrobial, and sensory attributes

**Rehenuma Tarannum[1], Asraful Alam[2], Md. Sakib Hasan[1], Golam Rabby[1], Ananya Raiyan[1], Rashida Parvin[1], Mahfujul Alam[2], Md. Ashrafuzzaman Zahid ●[1] ***

1 Department of Nutrition and Food Technology, Jashore University of Science and Technology, Jashore, Bangladesh, 2 Department of Agro Product Processing Technology, Jashore University of Science and Technology, Jashore, Bangladesh

* ashraf@just.edu.bd, zahid_24jstu@yahoo.com

**Data Availability Statement:** All relevant data are within the manuscript and its Supporting information files. We confirm that we've submitted

## Abstract

The objective of this study was the develop of fortified cookies enriched with oats flour and bitter gourd powder and monitoring the effects of these enrichments on the physicochemical, antioxidant, antimicrobial, and sensory attributes. This study was subjected to four treatments: control (0% oats flour and bitter gourd powder), T1 (10% oats flour), T2 (3% bitter gourd powder), and T3 (7% oats flour and 3% bitter gourd powder). Various physical properties of the cookies, including weight, thickness, diameter, spread ratio, baking loss, pH, and color values (L*, a*, and b*), were measured. Proximate analysis revealed moisture (4.23–4.70%), ash (1.17–1.67%), fat (13.62–15.09%), crude protein (7.02–7.36%), carbohydrate (71.78–72.97%), energy (442.62–452.40 kcal), and crude fiber (8.02–3.33%). Mineral contents included Na (787–754 mg/100g), Ca (873–435 mg/100g), and Zn (66.7–58.8 mg/100g). Additionally, DPPH free radical scavenging activity ranged from 13.14 to 75.51%, while TBARS activity varied from 0.78 to 1.33 mg MDA/kg. T2 cookies exhibited the highest antimicrobial activity, with control cookies showing the lowest. The 5-point hedonic scale indicated that T2 cookies had lower overall acceptability, while T3 cookies were better received. In conclusion, the study suggests that fortified cookies have a more significant impact than regular or control cookies.

## 1. Introduction

Cookies are a popular bakery product that is sweet in taste, small-shaped, and slightly raised; sometimes they can also be flat [1]. Cookies are widely recognized as a convenient and beneficial dietary source for the people. It contains vital nutrients, including macronutrients (carbohydrates, protein, and fat) and micronutrients (minerals and vitamins) [2].

Most bakery products primarily contain wheat flour (*Triticum aestivum*) as their key ingredient, serving as a major source of energy, carbohydrates, dietary protein, and a range of B

statistically analyzed data, and the dataset includes values along with means and standard deviations.

**Funding:** The author(s) received no specific funding for this work.

**Competing interests:** The authors have declared that no competing interests exist.

vitamins (thiamin, riboflavin, niacin, folate, and pantothenic acid). Wheat flour is the main component of almost all cookies; nevertheless, good quality products can be manufactured using non-wheat (nongluten) flours [3].

Oats (*Avena sativa L.*) are herbaceous biennials of the Poaceae family of plants [4]. Oats have traditionally been considered nutritious since they are high in protein, fiber, vitamins, and minerals and are primarily ingested in the form of oat meal. Oats are a high-value crop due to their high protein and fat content, balanced concentrations of important amino acids like lysine, and 2–6% β-glucan [5]. Oats contain about 95% fat in the form of palmitic, oleic, and linoleic acids, and 75–80% the remaining fat is unsaturated fatty acids. These unsaturated fatty acids are linked to a number of advantageous physiological characteristics, including antioxidant activity and the prevention of dementia [5–7]. Not only these nutrients but also polyphenols, tocopherols, vanillic acid, and avenanthramides are important antioxidants found in oats. Oats include bioactive compounds that have been demonstrated to enhance cholesterol, regulate blood sugar, protect against free radical damage, and support digestive health, among other health benefits. Oat flour is a form of flour prepared by grinding oats. It is a popular substitute to wheat flour for persons with gluten intolerance or celiac disease [8]. Bitter gourd (*Momordica charantia*), an annual vegetable, has been used as a functional food, nutraceutical, or herbal medicine since ancient times [9]. Bitter gourd is full of nutrients and a good source of vitamin A, vitamin B complex (thiamin, riboflavin, niacin, and folic acid), crucial amino acids (rich in seeds), minerals (macro-minerals: sodium, potassium, calcium, magnesium, and phosphorus; micro-minerals: iron, copper, zinc, and manganese), high fat content (seeds), and dietary fiber [10, 11]. Consuming bitter gourds in various forms, such as whole fruit, juice, extract, and dry powder, has been shown in studies to lower blood glucose levels. Bitter gourd contains health-beneficial flavonoids like ß-carotene, α-carotene, lutein, and zeaxanthin. It also contains vitamin A, which protects against ROS and free radicals that cause aging, cancer, and other diseases [12]. Moreover, it prevents atherosclerosis by regulating blood cholesterol [13].

To achieve optimal product quality in the industrial production of fiber-fortified cookies, it is essential to understand the functions of both ingredient and process variables while processing a food product, like cookies [14]. New technologies and ingredients are being implemented globally to meet nutritional needs. Functional foods, such as whole foods and fortified, enriched, or improved meals or dietary components, have the potential to reduce the risk of chronic disease and provide health benefits in addition to basic nutrition [15].

Wheat flour is commonly used as the principal ingredient in cookies recipes. A few studies have been done with the purpose of creating food products out of bitter gourd powder. However, the effects of oat flour and bitter gourd powder on the physicochemical, antioxidant, antimicrobial and sensory properties of fortified cookies have not yet been investigated. In this case, investigations were conducted on fortifying cookies with bitter gourd powder and oats flour. Furthermore, the cookies' physicochemical, antioxidant, antimicrobial and sensory properties were determined. Therefore, scientific evidence regarding product formulation, recipe suitability, and the impact of adding oats flour and bitter gourd powder on the nutritional content, sensory appeal, and quality of cookies is warranted to support the processing industry's use of oats flour and bitter gourd powder in for health and nutritional benefits. Consequently, the addition of these beneficial substances was intended to improve the nutritional advantages of the cookies, presenting an opportunity to identify an untapped market with high commercial potential.

## 2. Materials and methods

### 2.1 Schematic overview of the experimental program

The present study was focused on the development of oats flour and bitter gourd fortified cookies along with effects on physicochemical, antioxidant, antimicrobial, and sensory attributes. The experiment was conducted in the laboratory of the Department of Nutrition and Food Technology at Jashore University of Science and Technology, Jashore, Bangladesh. In this experiment, laboratory procedures were caried out in accordance with established standards. All chemicals and reagents used in the study were of analytical grade.

### 2.2 Raw materials for cookies preparation

Wheat flour (Pure all-purpose wheat flour, Bangladesh), oats (Kellogg's oats, India), bitter gourd and cooking salt (ACI salt, Bangladesh), soybean oil, white sugar, baking powder, and vanilla essence were procured from the local market in Jashore in order to prepare cookies. All ingredients were kept well in the Department of Nutrition & Food Technology's food research lab at Jashore University of Science and Technology.

### 2.3 Preparation of bitter gourd powder

The undesirable dirt of the bitter gourd was washed with normal temperature water and cut off with the proper thickness as it could dry well to obtain bitter gourd powder. The drying temperature was 55˚C at 48 h. At this temperature, antioxidant and nutrient values remained optimal. Using a grinder (Hotel King, Jaipan, India), the dried bitter gourd was ground to obtain bitter gourd powder. After sifting, the powder was kept at room temperature in an airtight glass container for further examination.

### 2.4 Formulation and preparation of cookies

The cookies were prepared with a basic formulation (Table 1), and several treatments were conducted by using different proportions of oat flour and constant portion bitter gourd powder incorporated with all-purpose wheat flour. Nevertheless, a control group was formulated solely using all-purpose wheat flour (constituting 65%). Additionally, three treatments were created to assess the characteristics of the final product. The first treatment involved using 55% all-purpose wheat flour and 10% oat flour; the second treatment utilized 62% all-purpose

Table 1. Formulation of oats flour and bitter gourd powder fortified cookies.

| Ingredients | Sample | | | |
|---|---|---|---|---|
| | Control | T1 | T2 | T3 |
| All-purpose wheat flour (g) | 65 | 55 | 62 | 55 |
| Oat flour (g) | 0 | 10 | 0 | 7 |
| Bitter gourd powder (g) | 0 | 0 | 3 | 3 |
| Sugar powder (g) | 16 | 16 | 16 | 16 |
| Baking powder (g) | 1.5 | 1.5 | 1.5 | 1.5 |
| Salt (g) | 0.5 | 0.5 | 0.5 | 0.5 |
| Vanilla essence (ml) | 2 | 2 | 2 | 2 |
| Oil (ml) | 15 | 15 | 15 | 15 |

[C = Control sample with solely 65% all-purpose wheat flour; T1 = Treatment-1 with 55% all- purpose wheat flour and 10% oats flour; T2 = Treatment-2 with 62% all-purpose wheat flour, and 3% bitter gourd powder,
T3 = Treatment-3 with 55% all-purpose wheat flour, 7% oats flour and 3% bitter gourd powder].

wheat flour and 3% bitter gourd powder; and the third treatment incorporated 55% all-purpose wheat flour, 7% oat flour, and 3% bitter gourd powder. The total weight of all the ingredients in cookies amounted to 100 grams.

After weighing all the ingredients, the all-purpose wheat flour and other dry ingredients were combined to achieve a consistent mixture. Vegetable oil and sugar powder were thoroughly mixed for approximately 2 minutes. Subsequently, blended dry ingredients were added slowly to the blended vegetable oil and fine sugar in the bowl, along with the addition of water (24 ml). The mixture was kneaded for approximately 3 minutes to form a soft dough. The dough was manually prepared by kneading. Once the dough reached the desired consistency, it was sealed with plastic wrap and allowed to rest for 10–15 minutes. Then, the resting dough was rolled out with a rolling pin on the wooden bread maker, and the round cookie shape was given to it. A baking sheet was used on the tray, and the oil was brushed on the baking sheet (making it easy to put on the prepared cookies). Cookies were placed on a greased baking sheet, placed in an oven that had been preheated to 160˚C, and baked for 25–30 minutes until the color of the cookies changed to a light brown color. Following the baking process, the cookies were promptly cooled, left at room temperature, and then sealed in polyethylene bags resistant to moisture and air for subsequent analysis (Fig 1) [16].

Measured of ingredients according to the prepared formulation

↓

All the dry ingredients were mixed well (except oil and fine sugar)

↓

Slowly added all ingredients to combined oil and fine sugar bowl and added water (24 ml) as needed to form soft dough

↓

Left the dough resting for 10-15 minutes and sealed with plastic wrap

↓

Rolled out the resting dough on a plain surface and provided desired shape to cookies

↓

The cookies were placed on greased (vegetable oil) baking sheet and put in the baking tray

↓

Baked at 160°C for 25-30 minutes

↓

Allowed the baked cookies to cool to room temperature

↓

Stored in air-resistant polyethylene bags at room temperature

**Fig 1. Schematic flow chart of developed cookies preparation.**

## 2.5 Physical properties of developed cookies

The weight (g), thickness (mm), diameter (mm), spread ratio, baking loss (%) and p$^H$ were assessed using three randomly chosen cookies from each sample. Cookie weight was determined by weighing three cookies from each sample and recording the average. Thickness and diameter were measured at two different places on each cookie using a vernier calliper, and the average was computed for each cookie. The spread ratio of cookies was computed by dividing the average diameter value by the average thickness value, following the method outlined by Emelike et al. [17]. The baking loss for cookies was determined by weighing three cookies before and after baking. The average difference in weight was calculated and resulted in the percentage of baking loss with minor modification [18]. For the pH test, 3 g of ground cookies sample underwent homogenization with 27 mL of distilled water for 30 seconds, utilizing a homogenizer. Subsequently, the slurry was allowed to settle at room temperature. The pH meter equipped with an electrode probe was calibrated using pH 7.00 buffer solutions. Readings were then taken three times for each treatment [19].

$$\text{Spread ratio} = \frac{\text{Average diameter}}{\text{Average thickness}} \tag{1}$$

$$\text{Baking loss (\%)} = \frac{\text{weight of before baking} - \text{weight of after baking}}{\text{Weight of before baking}} \times 100 \tag{2}$$

**2.5.1 Color of developed cookies.** The cookie samples color parameters were taken, followed by the method [20] by Minolta CR 300 colorimeter (Minolta, Tokyo, Japan). The L* values indicated lightness to darkness, a* value indicated the color greenness to redness and b* value indicated yellowness to blueness. Prior to measuring, the instrument was calibrated using the white paper for standardization.

## 2.6 Chemical analysis of developed cookies

**2.6.1 Determination of moisture.** The moisture content of the cookie samples was determined following the AOAC (2005) [21] method. Empty and clean petri dishes were subjected to drying in a dry oven (Ecocell, MMM Group, Germany) at 105˚C for 4 hours and subsequently placed in a desiccator for cooling. A precision balance machine (DJ-200AB, Shinko Denshi Co. Ltd., Japan) was used to weigh the empty petri dish and roughly 5 g of each sample. The samples were equally distributed in a petri dish before being dried in the dry oven for 4 hours at 105˚ C. After drying, the petri dish was moved to the desiccator for cooling, and the reweighing of the samples was carried out.

$$\text{Moisture(\%)} = \frac{\text{Weight of sample before drying} - \text{Weight of sample after drying}}{\text{Weight of sample before drying}} \times 100 \tag{3}$$

**2.6.2 Determination of fat.** The fat content of the cookie samples was determined by using AOAC (2005) [21] method with slight modifications. Ground samples totaling 5 g, along with 30 mL of n-hexane, were placed in a Soxhlet flask (SXT-06, Soxhelt Extraction Apparatus Set). The extraction of fat was conducted using n-hexane for a duration of 3 hours at 78˚C. Following the extraction, the sample solution was filtered through Whatman filter paper and transferred to a beaker. The filtrate was then cooked on a hot plate within the beaker. The beaker was cooled in a desiccator following the n-hexane evaporation, and the fat

content was then determined.

$$\text{Fat}(\%) = \frac{\text{Weight of the beaker with fat} - \text{Weight of the empty beaker}}{\text{Weight of sample}} \times 100 \qquad (4)$$

**2.6.3 Determination of protein.** The protein content of the cookie sample was determined using the AOAC (2005) [21] method with minor modification. In a cleaned and dried digestion tube, 1 g of powdered sample was combined with 2 g of digestion mixture ($K_2SO_4$ and Cu) and 20 mL of concentrated $H_2SO_4$. This mixture underwent digestion in a digester unit (1-BKD-20N, Biobase) through continuous heating at 450˚C for 2 hours until fumes became visible. Following digestion, the mixture was allowed to cool to room temperature before being diluted with 100 mL of distilled water. The sample was then diluted to 10 mL, combined with 50 mL of 40% NaOH solution, and distilled for 5 minutes in a distillation apparatus (BKN-984, Biobase). The filtrate was collected in a conical flask holding 25 mL of a 4% boric acid solution after the distillation. In order to finish the analysis, 1–2 drops of methyl red were added to the conical flask along with 9–10 drops of a mixed indicator. Then, titration against 0.1 N HCl was carried out until the color changed from pink to yellow. For the blank sample, which had no sample added, the identical process was carried out. Finally, 6.25 conversion factor was used to calculate total crude protein in cookies sample.

$$\text{Protein}(\%) = \frac{(\text{Titration value of sample} - \text{Titration value of blank}) \times \text{Normality of HCl} \times 0.014 \times \text{DF} \times 6.25}{\text{Sample weight}} \times 100 \quad (5)$$

**2.6.4 Determination of ash.** The ash content of the cookies sample was determined following the AOAC (2005) [21] method with minor modification. Approximately 5 g of each sample was placed in a dried and pre-weighed crucible, which was heated over a low Bunsen flame with the lid on. After the vapors had subsided, the crucible was placed in a box-type resistance muffle furnace (Shanghai Shuli Instruments & Meters Co., Ltd.) and heated to 600˚C for 5 hours. After the specified time, the crucible was transferred to a desiccator to cool, and its weight was measured again.

$$\text{Ash}(\%) = \frac{\text{Weight of crucible with final sample} - \text{Weight of empty crucible}}{\text{Weight of sample}} \times 100 \qquad (6)$$

**2.6.5 Determination of carbohydrate and energy.** The total carbohydrate of cookie sample was determined following the AOAC (2005) [21] method. The total percentage of Carbohydrates was computed after computing the previous nutrient content (percentage of moisture, fat, ash, and protein) in the sample.

$$\text{Total Carbohydrate}(\%) = 100 - (\text{moisture} + \text{protein} + \text{fat} + \text{ash}) \qquad (7)$$

The energy of cookie sample was calculated following AOAC (2005) [21] method. After computing carbohydrate, protein, and fat, the total calorie was calculated.

$$\text{Total Calorie} = 4 \times \text{Carbohydrate}(\%) + 4 \times \text{Protein}(\%) + 9 \times \text{Fat}(\%) \qquad (8)$$

**2.6.6 Determination of fiber.** The fiber content of the cookie sample was assessed using the previous work, with minor modifications mentioned by Hamid and Hamid [22]. A beaker was filled with approximately 10g of moisture- and fat-free ground cookies. Then, 200 ml of

1.25% H2SO4 solution was added to the same beaker of sample, and the mixture was heated for 30 minutes on a hot plate (78–1 Magnetic Stirrer Hotplate, China) with intermittent stirring. After boiling, the solution was chilled and filtered through muslin fabric. The residue on the cloth was then transferred back to the beaker with a spatula and boiled for another 30 minutes, this time in 200 mL of 1.25% NaOH solution. After cooling, the sample was filtered again using a muslin towel and the residue was washed with hot water. The remaining sample was then placed in a crucible and dried overnight in a drier set to 105˚C. A desiccator was used to cool the crucible before it was weighed. The crucible was then placed in a muffle furnace and heated for three hours at 600˚C. After that, the crucible was placed to the desiccator, let to cool, and then weighed again.

$$Fiber(\%) = \frac{\text{Dried sample weight} - \text{Ash weight of sample}}{\text{Initial sample weight}} \times 100 \tag{9}$$

**2.6.7 Determination of minerals (Na, Ca, and Zn).** The wet digestion of materials was carried out using the approach detailed by Yilmaz and Karaman [23], with minor modifications. Initially, 1 g of ground cookies was measured and placed in a 100-mL beaker. Subsequently, 15 mL of 65% HNO3 was added to the beaker containing the sample and left undisturbed for 24 hours, covered. The following day, the dissolved sample was heated on a hot plate at 120–160˚C until the solution volume was decreased to one-third of its original. Following this, 5 mL of deionized water was added to the solution and boiled again. Finally, 5 mL of H2O2 was added, and the solution was heated to one-third of its original volume. The solution was then allowed to cool and filtered using white Whatman qualitative filter papers (125 mm diameter). Subsequently, the filtered solutions were diluted with deionized water up to the 100 mL mark in a volumetric flask. The diluted solutions were then transferred into reagent bottles and stored in a refrigerator.

**2.6.8 DPPH free radical scavenging activity of developed cookies.** The assessment of DPPH (2,2-diphenyl-1-picrylhydrazyl) radical scavenging activity followed the procedure outlined by Alam et al. [24] with some adaptations. A solution of 24mM DPPH in methanol (2,850μL) was combined with 150μL of the sample, and the mixture was allowed to react in darkness for 24 hours. The absorbance was measured at 515 nm using a spectrophotometer. To calculate DPPH radical scavenging activity, the percentage difference (%) between the sample's absorbance was determined.

$$Antioxidant\ activity(\%) = \left(1 - \frac{\text{Absorbance of sample}}{\text{Absorbance of control}}\right) \times 100 \tag{10}$$

**2.6.9 TBARS of developed cookies.** In order to perform TBARS (Thiobarbituric Acid-Reactive Substances) analysis on the cookies sample, 3 g of ground samples were homogenized at a high speed using a digital homogenizer for 20 seconds with 27 mL of 3.86% perchloric acid. The samples were then allowed to settle at a low temperature for an hour. The combination was then centrifuged for 10 minutes at 2000 rpm to separate the lipid component from the other macronutrients. The cleared solution was then produced by filtering it using Whatman 41 filter paper. Two milliliters of the 20 milliliter TBA solution and two milliliters of the filtered solution were pipetted into a test tube. Two milliliters of distilled water were mixed with two milliliters of the 20-mM TBA solution to form a blank sample. All solutions were then carefully stored at room temperature for 15 hours. Finally, the concentration of TBARS was determined using a spectrophotometer by measuring the absorbance at 531 nm. All

treatments were prepared in triplicate for analysis [25].

$$\text{TBARS} = (\text{Absorbance of sample} - \text{Absorbance of control}) \times 5.54 \tag{11}$$

## 2.7 Determination of total plate count of developed cookies

The diluent and culture media underwent autoclaving at 121°C for (1.5–2) h. 1 gram of each cookies sample was dissolved in 10 ml of sterile sodium chloride solution, thoroughly mixed by swirling, and then further diluted to achieve concentrations of $10^{-1}$, $10^{-2}$, $10^{-3}$, $10^{-4}$, $10^{-5}$, $10^{-6}$, and $10^{-7}$ using a fresh sterile pipette for each dilution. The microbial population was evaluated using the spread plate technique. Each petri dish was filled with 20 ml of sterile nutrient agar medium, allowed to cool for solidification, and then the last three diluted concentration $10^{-5}$, $10^{-6}$, $10^{-7}$ were used duplicate in petri dish for analysis, 0.1 mL of each dilution was transferred to the corresponding plates. Using a sterile glass spreader, the sample was evenly spread in circular movements in various directions for 10 seconds. Following inoculation, the plates were incubated at 37°C for 24 hours. Following incubation, the number of colonies was determined and expressed as colony-forming units (cfu/ml) per milliliter [26].

$$\text{Total plate count}(\text{cfu/ml}) = \frac{\text{Number of colonies counted} \times \text{Dilution factor}}{\text{Volume of cultured plated in ml}} \tag{12}$$

## 2.8 Sensory attributes of cookies

Cookie samples were prepared using all-purpose wheat flour, oats flour, and bitter gourd powder, and assigned distinct random codes. These samples were presented to each panelist in a randomized order. Four varieties of cookies were assessed individually. Participants (all of above 16 years age) agreed to participate in the sensory examination. The sensory evaluation followed the protocols approved by the ethical committee. Furthermore, the research was carried out in accordance with the standards provided by the Nutrition and Food Technology departmental ethical committee (Dr. Omar Faruque, Dr. Md. Alauddin and Dr. Md. Mahmudul Hasan) at Jashore University of Science and Technology, Jashore-7408, Bangladesh. A sensory panel consisting of 10 members evaluated sensory quality attributes, including hardness, appearance, chewiness/crispiness, flavor/color, and overall acceptability, using a 5-point Hedonic scale represented- mostly like (5), moderately like (4), not like nor dislike (3), moderately dislike (2), and mostly dislike (1) [1].

## 2.9 Statistical analysis of developed cookies

All the experiments were conducted in analysis of variance (ANOVA) to identify the significant difference in the mean value of the developed cookies. The Tukey's honestly significant difference test (Tukey's HSD) is used to test differences among sample means for significance. Significance was accepted at $p < 0.05$. The study was replicated three times to explain the source of variation precisely. In addition, total plate count of cookies conducted both in ANOVA and Independent T test (when replicated two times), simple logistic regression was used to draw the graphs of DPPH and TBARS. The analysis was performed with the software SPSS version 26.

# 3. Results and discussion

## 3.1 Weight, thickness, diameter, spread ratio, baking loss and pH

Table 2 shows the physical properties of developed cookies at different treatment (Control = 0% treated; T1 = 10% oats flour; T2 = 3% bitter gourd powder; T3 = 7% oats flour and 3% bitter

**Table 2. Weight, thickness, diameter, spread ratio, baking loss, and p^H of developed cookies at different treatment.**

| Parameters | Control | T1 | T2 | T3 |
|---|---|---|---|---|
| Weight (g) | 10.86 ± 1.51[a] | 9.91 ± 1.38[a] | 9.37 ± 1.15[a] | 10.84 ± 1.45[a] |
| Thickness (mm) | 13.54 ± 0.72[a] | 12.06 ± 0.60[ab] | 12.44 ± 0.96[ab] | 11.12 ± 0.46[b] |
| Diameter (mm) | 36.82 ± 3.13[a] | 38.56 ± 1.056[a] | 35.37 ± 1.38[a] | 41.08 ± 2.82[a] |
| Spread ratio | 2.72 ± 0.12[bc] | 3.20 ± 0.078[b] | 2.85 ± 0.23[bc] | 3.69 ± 0.26[a] |
| Baking loss (g/100) | 20.67 ± 1.54[a] | 22.15 ± 2.20[a] | 23.77 ± 1.76[a] | 22.12 ± 1.15[a] |
| p^H | 6.92 ± .015[a] | 6.83 ± .026[c] | 6.81 ± .010[ac] | 6.84 ± .026[bc] |

Note: Distinct superscript letters [(a-c)] within a raw indicate statistically significant and differing mean values among various (p<0.05) from each other. Post-hoc test outcomes were derived from Tukey's HSD test for cases of homogeneous variance. The provided values represent the means of three replicates with their corresponding standard deviations. (Control = without oats flour and bitter gourd powder; T1 = 10% oats flour; T2 = 3% bitter gourd powder; T3 = 7% oats flour and 3% bitter gourd powder).

gourd powder). The different dimensional physical properties such as weight (g), thickness (mm), diameter (mm), spread ratio, and baking loss (%) and p^H are demonstrated in this table.

The table indicates mean weight and diameter of difference cookies have no significant difference (p<0.05) among them that means weight and diameter does not depend on elements of the treatments in this study. In the case of thickness of control and T3, both of them shows significance relation to each other that means p<0.05. Besides T1 and T2 shows no significance with any rest of treatments. Thickness (mm) of the control sample significantly (p<0.05) higher than T3 sample. Gluten formation and fiber can influence the thickness of cookies. In the case of T3, the thickness is reduced compared to others due to excessive gluten development resulting from increased kneading time [27]. These findings are in line with an earlier investigation that showed adding apple pomace powder decreased the baked cookies' thickness [28].

The mean spread ratio of Control, T1, T2 have significant difference (p<0.05) with T3 but T1 has no significance relation with other treatments except control. The maximum and minimum spread ratio are found in T3 = 3.69 and control = 2.72, respectively. On the other hand, baking loss has no significance relation among treatments. The maximum and minimum baking loss are found in T2 (23.77%) and control (20.67%), respectively. The spread ratio depends on diameter and thickness of cookies but the thickness of cookies of T3 sample has already influenced by the gluten development. Besides, control spread ratio is less than other treatments because oats flour and bitter gourd powder are good source of fiber that has ability to absorb water and swells up in size. The diameter or volume of the cookies is usually dictated by the quantity and quality of flour proteins, especially gluten; when other fruit flours are combined with wheat flour, the protein content of the composite flour decreases, which also affects the spread ratio and causes a decrease in cookie volume [29].

The mean baking loss (%) has no significance difference among the treatments (p>0.05) where baking loss is supposed to removal of the moisture. The baking loss is not much different from each other (control = 20.67%, T1 = 22.15%, T2 = 23.77%, and T3 = 22.12%). This result shows the similar report with Mohibbullah et al. [30]. According to a recent study by Rafique et al. [31], adding non-wheat flour derived from fruits and vegetables to bakery products caused unwanted changes in their physical characteristics, possibly due to the disrupted the network and concentration of gluten. For this reason, when trying to create nutritious products, it's important to choose the right amount of replacement for these flours to minimize unintended changes.

The mean $p^H$ has significance relation ($p<0.05$) between control and other treatments. T1, T2, and T3 have only significant relation with control but not with others. The pH of control is 6.92 which is higher than $p^H$ of T1 (6.83), T2 (6.81), and T3 (6.84) are fluctuated a little bit with each other due to their % of elements and the presence of polysaccharides and phenolic compounds. However, this result is considerably similar to Oh et al. [32].

**3.1.1 Color value.** Color is a crucial characteristic for evaluating baking efficiency since it conveys information about product formulation, quality, and the ability of raw materials. For color analysis, the parameters for the color measurement (L*, a*, and b*) of the prepared cookies were utilized, as shown in Table 3. The oats flour with different percentage (T1 = 10%, and T3 = 7%) and bitter gourd with constant percentage (T2 = 3%, and T3 = 3%) are influenced the color properties of cookies. Here, L* value of Control represents significant difference ($p<0.05$) between T2 and T3. However, there was no significant difference between T1 and control of L*; even no significant difference between T2 and T3 of L*. Again, T3 showed significant difference between control and T1 of L* value. It is noticeable that the control cookies L* (Lightness) 69.06, significantly increased but it is decreased from T1 (68.80), T2 (59.28), and T3 (58.51) with added oats flour and bitter gourd powder. The findings indicate that cookies with a higher number of functional components, such as oat flour and bitter gourd powder, are darker than those with less functional ingredients. Similar effect was observed in various baked foods that contained functional compounds derived from fruit byproducts such as orange peel, apple pomace, and pineapple peel [33]. The cookies' dark color is mostly caused by the Maillard reaction, which happens between amino acids and reducing sugar at high temperatures during baking, as well as caramelization after the temperature hits 150˚C [34].

Redness (a*) value of control (1.69) shows significant difference ($p<0.05$) among T1 (1.52), T2 (-4.58) and T3 (-4.24). Even T1 represents significant difference among control, T2 and T3 of a* value. However, there was no significant difference between T2 and T3 of a* color value. The positive value of a* (Redness) indicates red color and negative value indicates green color, a* value is high in control comparison to other treatments. In T1, T2, and T3; a* value is decreased due to add oats flour and bitter gourd powder. The negative value of T2 and T3 shows the green color of cookies. Again, yellowness (b*) value shows there was significant difference ($p<0.05$) among control (14.39), T1(21.04), T2 (16.54), and T3 (24.02). b* (Yellowness) shows yellow color due to positive value and blue color due to negative value. There are significant difference ($p<0.05$) among the treatments because of the cookies elements according to Ashoush and Mahdy; Susanti et al. [18, 35].

**3.1.2 Moisture (%), protein (%), fat (%), ash (%), carbohydrate (%), and energy (kcal).** Table 4 represents the proximate composition of developed cookies at different treatments. In this study, there is no significant relation between the treatments of moisture content. According to Emelike et al. [17], The ideal cookies moisture content is less than 5% and all treatments'

**Table 3. Color value of developed cookies at different treatment.**

| Parameters | Control | T1 | T2 | T3 |
|---|---|---|---|---|
| Lightness (L*) | 69.06 ± 3.73[a] | 68.80 ± 1.89[a] | 59.28 ± 2.70[b] | 58.51 ± 2.17[b] |
| Redness (a*) | 1.69 ± .47[a] | 1.52 ± .37[b] | -4.58 ± .54[c] | -4.24 ± .65[c] |
| Yellowness (b*) | 14.39 ± .644[d] | 21.04 ± .56[b] | 16.54 ± .808[c] | 24.02 ± .408[a] |

Note: Distinct superscript letters ([a-d]) within a raw indicate statistically significant and differing mean values among various at ($p<0.05$) from each other. Post-hoc test outcomes were derived from Tukey's HSD test for cases of homogeneous variance. The provided values represent the means of three replicates with their corresponding standard deviations. (Control = without oats flour and bitter gourd powder; T1 = 10% oats flour; T2 = 3% bitter gourd powder; T3 = 7% oats flour and 3% bitter gourd powder).

**Table 4. Proximate composition of developed cookies (%).**

| Treatments | Control | T1 | T2 | T3 |
|---|---|---|---|---|
| Moisture (%) | 4.23 ± .115[a] | 4.53 ± .38[a] | 4.40 ± .30[a] | 4.70 ± .173[a] |
| Protein (%) | 7.32 ± .0252[a] | 7.36 ± .020[a] | 7.02 ± .003[b] | 7.06 ± .0153[b] |
| Fat (%) | 14.30 ± 1.514[a] | 15.09 ± .61[a] | 13.82 ± .92[a] | 13.62 ± 1.28[a] |
| Ash (%) | 1.17 ± 0.153[b] | 1.23 ± 0.115[b] | 1.50 ± 0.20[bc] | 1.67 ± 0.058[ac] |
| Carbohydrate (%) | 72.97 ± 1.49[a] | 71.78 ± .628[a] | 72.92 ± 1.005[a] | 72.96 ± 1.06[a] |
| Energy value (Kcal) | 449.91 ± 7.60[a] | 452.40 ± 3.51[a] | 444.18 ± 5.43[a] | 442.62 ± 7.23[a] |
| Crude fiber (%) | 3.33 ± 0.11[d] | 4.04 ± 0.20[c] | 6.01 ± 0.20[b] | 8.02 ± 0.20[a] |

Note: Distinct superscript letters ([a-d]) within a raw indicate statistically significant and differing mean values among various at (p<0.05) from each other. Post-hoc test outcomes were derived from Tukey's HSD test for cases of homogeneous variance. The provided values represent the means of three replicates with their corresponding standard deviations. (Control = without oats flour and bitter gourd powder; T1 = 10% oats flour; T2 = 3% bitter gourd powder; T3 = 7% oats flour and 3% bitter gourd powder).

moisture content in this study is less than 5%. A maximum moisture content of 5% in cookies can help to prevent physical damage such as cookies becoming slow and losing their crispiness, as well as damage caused by microorganisms like mold. The change in moisture content of the result can be produced by a variety of factors, including the water content of the raw material and the processing method.

In the case of protein (%) both control and T1 has significant relation with T2 and T3 but there is no relation with both of them. Control protein 7.32%, and T1 (10% oats flour) protein is increased 7.36%. But T2 protein 7.02% and T3 protein 7.06% significantly decreased due to add bitter gourd powder (3%) in T2 and combined oats flour (7%) and bitter gourd powder (3%) in T3 treatment because bitter gourd contains less amount of protein than oats flour reported by Bhuiyan et al. [36]. Protein contain in bitter gourd (*M. charantia* hybrid variety Gojnee) is 1.02% according to Bhuiyan et al. [36] and oats flour contain 11.61% according to Youssef et al. [37]. There is no significant difference among fat (%) of developed cookies. Control, T1, T2 and T3 contain fat content 14.30%, 15.09%, 13.82% and 13.62% respectably.

In the case of ash (%), control and T1 shows significant difference (p<0.05) with T3. And T2 was not significant with control, T1, and T3. Again, T3 showed significant difference with control and T1 but not significant with T2 in ash (%). Ash content pertains to the assessment of inorganic non-combustible materials, and the percentage of ash in T1 (1.23%), T2 (1.50%), and T3 (1.67%) has elevated compared to the control (1.17%). This increase is attributed to the inclusion of 10% oats flour in T1, 3% bitter gourd powder in T2, and a combination of 7% oats flour and 3% bitter gourd powder in T3. This result shows similar report with Ashoush and Mahdy; Susanti et al. [18, 35]. The ash content of a substance is strongly connected to its mineral content. With controlled circumstances and processing procedures, it is believed that the variance in ash content is due to the product's mineral content. Cookies treated with vegetable powder had a higher ash level than control cookies [38].

Carbohydrate serves as the primary macronutrient in the cookies. The study indicates that there is no significant difference in the percentage of carbohydrates among the treatments. Similarly, there is no significant difference observed in the energy values (kcal) among the treatments, as reported by Ashoush and Mahdy [18].

On the other hand, there is significant difference (p<0.05) among the treatments of fiber (%). The crude fiber (%) is increased T1 (4.04%), T2 (6.01%), and T3 (8.02%) compared to control (3.33%). The fiber content of the cookies increased when oats flour and bitter gourd powder were substituted for wheat flour because they had a significantly greater fiber content.

**Table 5. Mineral contents of developed cookies.**

| Treatments | Control | T1 | T2 | T3 |
|---|---|---|---|---|
| Na (mg/100g) | 771 ± 0.03[b] | 754 ± 0.03[d] | 763 ± 1.53[c] | 787 ± .006[a] |
| Ca (mg/100g) | 435 ± 0.006[d] | 599 ± 0.006[c] | 873 ± 0.006[a] | 807 ± 0.006[b] |
| Zn (mg/100g) | 58.8 ± 0.03[d] | 59.0 ± 0.06[c] | 66.7 ± 0.06[a] | 64.8 ± 0.025[b] |

Note: Distinct superscript letters ([a-d]) within a raw indicate statistically significant and differing mean values among various at (p<0.05) from each other. Post-hoc test outcomes were derived from Tukey's HSD test for cases of homogeneous variance. The provided values represent the means of three replicates with their corresponding standard deviations. (Control = without oats flour and bitter gourd powder; T1 = 10% oats flour; T2 = 3% bitter gourd powder; T3 = 7% oats flour and 3% bitter gourd powder).

These results are in line with those of Kausar et al. [29], who discovered that cookies containing higher amounts of grapefruit pomace powder had higher fiber contents. Based on this proximate data, it is possible to conclude that cookies have higher ash, fat, protein, carbohydrate, and fiber levels than control cookies, indicating that cookies have strong nutritional value and the potential to be developed further.

**3.1.3 Minerals content (Na, Ca, and Zn).** Table 5 displays the mineral contents of Na (mg/100g), Ca (mg/100g), and Zn (mg/100g). The table highlights significant differences (p<0.05) among control, T1, T2, and T3 in minerals (Na, Ca, and Zn).

Na content in the control cookies is measured at 771 mg/100g. In T1, the Na level is lower than the control, measuring 754 mg/100g. T2 contains 763 mg/100g of Na, also indicating a lower amount than the control. However, in T3, the Na content increases to 787 mg/100g, surpassing that of the control. The use of 10% oats flour in T1 results in a significantly decreased Na level (244 mg/100g), aligning with Rybicka and Gliszczyńska-Świgło [39] findings where 10% oats flour, combined with gluten-free flour, led to reduced Na content. Conversely, the increase in Na levels in T2 and T3 is attributed to the noticeable sodium percentage in bitter gourd powder, as reported by Krishnendu and Nandini [40] highlighted the diverse nutritional composition of various types of bitter gourd with dark green small bitter gourd containing 4.37 mg/100g Na.

The calcium (Ca) levels have shown an increase in T1 (599 mg/100g), T2 (873 mg/100g), and T3 (807 mg/100g) compared to the control (435 mg/100g). This rise can be attributed to the varying Ca levels in the ingredients, with all-purpose wheat flour (white flour) containing 2.80 mg/100g, oats flour (yellow variety) possessing a Ca level of 44 mg/g, and bitter gourd (dark green small) having 22.79 mg/100g, as documented by Krishnendu and Nandini; Alemayehu et al. [40, 41]. All-purpose wheat flour contains a lesser amount of calcium compared to other elements. Consequently, the addition of oat flour in T1, bitter gourd powder in T2, and the combination of oat flour and bitter gourd powder in T3 results in an increased Ca level. However, the Ca level in T3 decreases compared to T2 due to the lower percentage of oat flour in the combined mixture. Gebremariam et al. [42] studied calcium content of cookies supplemented with pumpkin seed flour, which is higher than control cookies made with wheat flour.

The Zinc (Zn) levels have shown an increase in T1 (59.0 mg/100g), T2 (66.7 mg/100g), and T3 (64.8 mg/100g) compared to the control (58.8 mg/100g). This rise can be attributed to the varying zinc levels in the ingredients, with all-purpose wheat flour (white flour) containing 0.61 mg/100g, oats flour (yellow variety) possessing a Zn level of 2 mg/100g, and bitter gourd (dark green small) having 70.14 mg/100g, as documented by Krishnendu and Nandini; Alemayehu et al. [40, 41]. All-purpose wheat flour contains a lesser amount of zinc compared to other elements. Consequently, the addition of oat flour in T1, bitter gourd powder in T2, and

the combination of oat flour and bitter gourd powder in T3 results in an increased zinc level. However, the zinc level in T3 decreases compared to T2 due to the lower percentage of oat flour in the combined mixture. Mineral concentrations in vegetables can vary based on the intrinsic (varieties, maturity, genetics, and age) and environmental (soils, geographical regions, season, water source, and fertilizer use) characteristics of plants and animals, as well as the ways of handling and processing [38].

**3.1.4 DPPH, TBARS, and anti-microbial activity.**   Antioxidants are considered to play an important effect in preventing oxidative cell damage [43]. Fig 2 shows the changes in developed cookies (Control = without oats flour and bitter gourd powder; T1 = 10% oats flour; T2 = 3% bitter gourd powder; T3 = 7% oats flour and 3% bitter gourd powder) due to effect of DPPH free radical scavenging activity. Three replicated mean values are taken and error bars show the standard error. Cookies fortified with oat flour and bitter gourd powder exhibited enhanced free radical scavenging activity compared to the control treatment. Oats and bitter gourd are recognized for their antioxidant properties. The mean DPPH (%) values were recorded as follows: control (13.14), T1 (22.42), T2 (75.51), and T3 (54.69). Fig 2 illustrates that T2 (3% bitter gourd) exhibited the highest value, while the control showed the lowest. However, T3 demonstrated lower DPPH (%) than T2, attributable to the variation in oat flour percentage. Our findings show that replacing flour with oat flour and bitter gourd powder dramatically improved antioxidant activity. Several researches have revealed similar findings about the increased antioxidant activity of composite bakery goods, including cookies. An increase in antioxidant activity in cookies after baking, possibly related to the synthesis of brown color compounds during baking (Maillard reaction). A similar pattern of antioxidant activity appeared in the cookies that included Manisha Jose pineapple pomace and *Tinospora cordifolia* (TC) stem powder [44]. Similar outcomes, demonstrating an increase in RWF replacement with PPD to enhance the antioxidant content of cookies, were also demonstrated by Chagas et al. [45].

Fig 3 shows the changes in developed cookies (Control = without oats flour and bitter gourd powder; T1 = 10% oats flour; T2 = 3% bitter gourd powder; T3 = 7% oats flour and 3% bitter gourd powder) due to effect of TBARS. Three replicated mean values are taken and

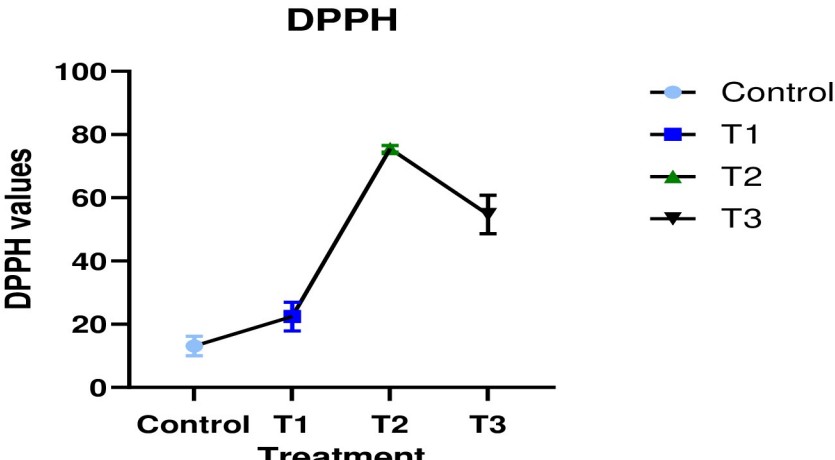

**Fig 2. Effect of DPPH free radical scavenging activity on developed cookies (Control = without oats flour and bitter gourd powder; T1 = 10% oats flour; T2 = 3% bitter gourd powder; T3 = 7% oats flour and 3% bitter gourd powder) due to effect of DPPH free radical scavenging activity.** Three replicated mean values are taken and error bars show the standard error.

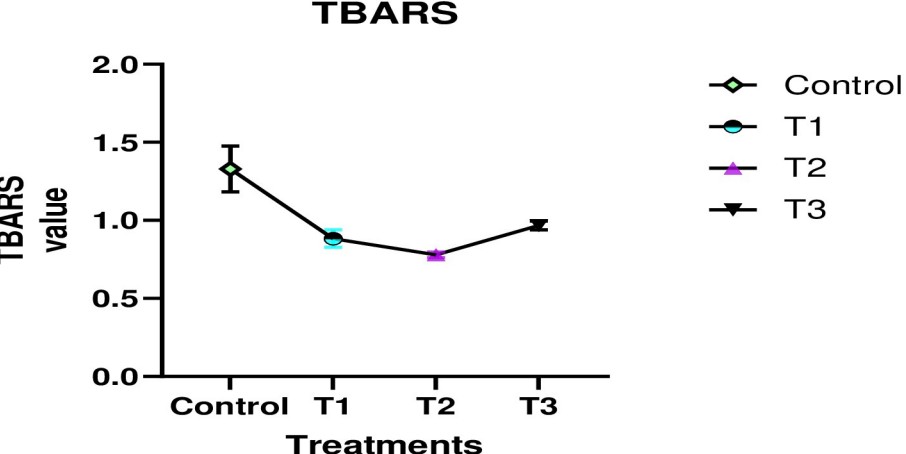

**Fig 3. Effect of TBARS values on developed cookies (Control = without oats flour and bitter gourd powder; T1 = 10% oats flour; T2 = 3% bitter gourd powder; T3 = 7% oats flour and 3% bitter gourd powder) due to effect of TBARS.** Three replicated mean values are taken and error bars show the standard error.

error bars show the standard error. The TBARS (Thiobarbituric Acid Reactive Substances) assay was employed to assess lipid peroxidation in the developed cookies. The cookies are rich in antioxidants, which contribute to reducing oxidative stress on lipids. In this assay, TBA reacts with the byproduct of lipid peroxidation, malondialdehyde, forming a colored complex. The intensity of the color is indicative of the extent of lipid peroxidation in the cookies. Fig 3 illustrates the lowest TBA value (T2 = 0.78 mg MDA/kg) and the highest TBA value (control = 1.33 mg MDA/kg). Furthermore, T3 (0.9676 mg MDA/kg) exhibits an increased TBA value compared to both T1 (0.88 mg MDA/kg) and T2 (0.78 mg MDA/kg), attributable to the variation in oat percentage.

The findings of this study show that including oat flour and bitter gourd powder in cookies efficiently reduces lipid oxidation, similar to the performance of the synthetic antioxidant TBHQ. This result is consistent with prior research, which has demonstrated that natural antioxidants, such as phenolic compounds, play an important role in preventing oxidative damage in baked goods. Lyophilized peach on yam cookies can sustain reduced TBARS values, which contributes to their stability [46].

Table 6 displays the antimicrobial activity of the developed cookies assessed using the total plate count method. The microbial growth on day-01 exhibits no significant difference

**Table 6. Antimicrobial activity of developed cookies.**

| Treatment | Control | T1 | T2 | T3 |
|---|---|---|---|---|
| Day-01 | 7.40 ±.175[Aa] | 7.46 ± .151[Aa] | 7.10 ± .175[Aa] | 7.26 ± .241[Aa] |
| Day-25 | 8.30 ± .128[Ba] | 8.21 ± .040[Ba] | 8.00 ± .053[Bb] | 8.16 ± .017[Ba] |

Note: Distinct superscript letters ([a-b]) within a raw indicate statistically significant and differing mean values among various at ($p < 0.05$) from each other. Post-hoc test outcomes were derived from Tukey's HSD test for cases of homogeneous variance and independent samples t-test for compare means of two unrelated groups indicate superscript letter ([A-B]). The provided values represent the means of three replicates with their corresponding standard deviations. (Control = without oats flour and bitter gourd powder; T1 = 10% oats flour; T2 = 3% bitter gourd powder; T3 = 7% oats flour and 3% bitter gourd powder).

(p<0.05) among the treatments. However, on day-25, the antimicrobial activity demonstrates a significant difference, particularly between T2 and the other treatments. Additionally, T3 also exhibits an impact on antimicrobial activity, with a count marginally significant or lower than that of the control and T1. Therefore, bitter gourd has antimicrobial effects on the developed cookies and it can extend the shelf life and quality of cookies which is agreed with Mohite and Waghmare [26]. Moreover, compare with day-01 and day-25, there was significant difference (p<0.05) in microbial growth.

**3.1.5 Sensory attributes.** Sensory evaluation is often performed at the end of product formulation to see whether a new product will appeal to customers. Cookies prepared with oat flour and bitter gourd powder were tested for sensory properties, as shown in Fig 4. The developed cookies hardness shows there was no significant difference (p>0.05) among control, T1, T2, and T3. However, as more oat flour and bitter gourd powder were added to the formulation, the cookies' hardness increased. similarly, it has been documented that incorporating HPB (hog plum bagasse) into composite cookies greatly enhanced their hardness [47].

Cookie's appearance, crispiness, and flavor of T2 showed significant difference with control, T1, and T3. While overall acceptability dictates there was significant difference in control with T3, besides T1 and T2 were not significant (p>0.05) but T1 and T2 were significant (p<0.05) with control and T3. The incorporation of 62% wheat flour and 3% bitter gourd powder in T2 results in a bitter taste and a light green color, influencing the acceptability. Conversely, T3 cookies, with 7% oat flour and 3% bitter gourd powder, exhibit higher acceptability than T2 due to reduced bitterness. The interaction between the fiber-rich powder and the macromolecules in the wheat dough may be the cause of the observed differences in the sensory attributes of the generated cookies [32]. According to a similar study, adding 0.5% of clove powder [48] and 8% of *Tinospora cordifolia* stem powder [15] to cookie recipes produced satisfactory results without degrading the cookies' sensory qualities. The addition of any ingredient

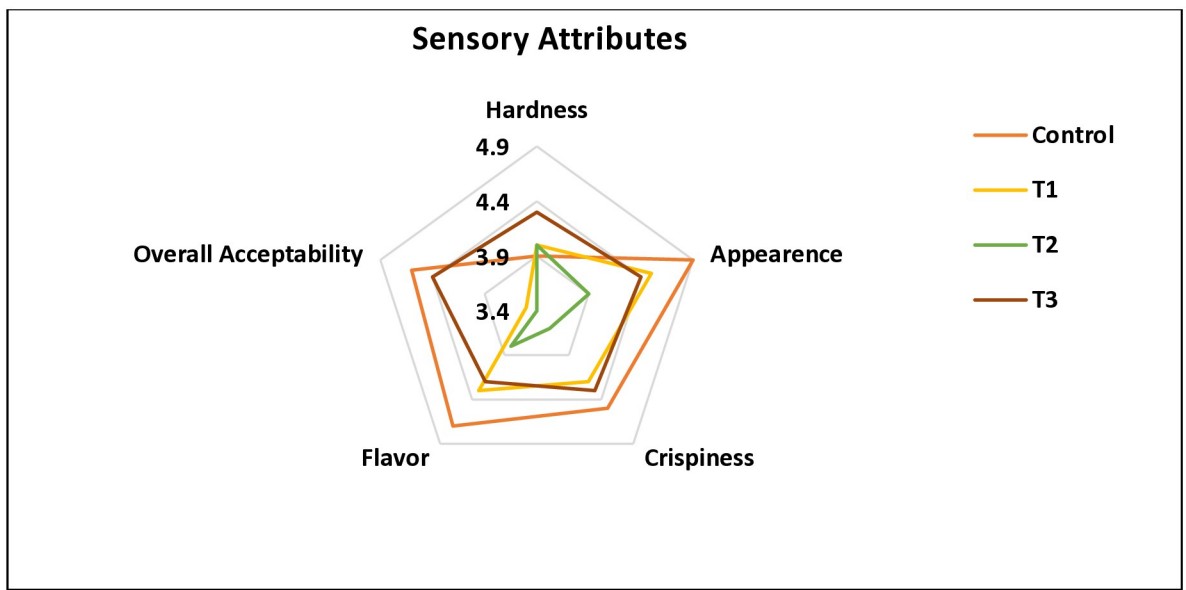

**Fig 4. Sensory attributes of developed cookies.** Note: Distinct superscript letters (a-b) within a raw indicate statistically significant and differing mean values among various at (p<0.05) from each other. Post-hoc test outcomes were derived from Tukey's HSD test for cases of homogeneous variance. The provided values represent the means of three replicates with their corresponding standard deviations. (Control = without oats flour and bitter gourd powder; T1 = 10% oats flour; T2 = 3% bitter gourd powder; T3 = 7% oats flour and 3% bitter gourd powder).

to cookies alters their texture, flavor, color, and taste [49]. From the current finding, cookie samples in T3 were found most acceptable, therefore a 7% oat flour and 3% bitter gourd powder could be a suitable level of replacement with wheat flour to develop acceptable bakery items.

## 4. Conclusions

This cookies product development study shows the fortified cookies are elevated in their physicochemical properties, antioxidant levels, and antimicrobial effects. Even the sensory attributes of fortified cookies are noticeable where T3 cookies have more acceptability than other fortified treatments. Moreover, the hypothesis is that the developed cookies will provide potential benefits for individuals dealing with conditions like diabetes, high blood pressure, and heart disease. There is a call to promote the consumption of oats and bitter gourd as functional and nutraceutical foods, urging the food industry to innovate and develop such inventive dietary supplement sources for bakery products. However, further studies are required to accurately determine parameters such as phenolic content, flavonoids, other antioxidant activities, and the shelf life of these cookies.

## Supporting information

**S1 Dataset.**
(DOCX)

## Acknowledgments

The authors acknowledge the kind support from the department of Nutrition and Food Technology and the department of Agro Product Processing Technology of Jashore University of Science and Technology, Jashore-7408, Bangladesh.

## Author Contributions

**Conceptualization:** Rehenuma Tarannum, Md. Ashrafuzzaman Zahid.

**Data curation:** Rehenuma Tarannum, Md. Sakib Hasan, Golam Rabby, Ananya Raiyan.

**Formal analysis:** Rehenuma Tarannum, Md. Sakib Hasan, Golam Rabby, Ananya Raiyan.

**Funding acquisition:** Md. Ashrafuzzaman Zahid.

**Investigation:** Md. Ashrafuzzaman Zahid.

**Methodology:** Rehenuma Tarannum.

**Resources:** Md. Ashrafuzzaman Zahid.

**Software:** Rehenuma Tarannum.

**Supervision:** Md. Ashrafuzzaman Zahid.

**Validation:** Asraful Alam.

**Visualization:** Asraful Alam.

**Writing – original draft:** Rehenuma Tarannum, Md. Sakib Hasan, Golam Rabby.

**Writing – review & editing:** Asraful Alam, Ananya Raiyan, Rashida Parvin, Mahfujul Alam.

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
