## [Decision Letter · Decision Letter 0]

18 Jul 2024

PONE-D-24-26098Development of oats flour and bitter gourd fortified cookies: effects on physicochemical, antioxidant, antimicrobial, and sensory attributesPLOS ONE

Dear Dr. Zahid,

Thank you for submitting your manuscript to PLOS ONE. After careful consideration, we feel that it has merit but does not fully meet PLOS ONE’s publication criteria as it currently stands. Therefore, we invite you to submit a revised version of the manuscript that addresses the points raised during the review process.

**Please check comments from all reviewers and revise the paper carefully. All responses should be provided. ** 

We look forward to receiving your revised manuscript.

Kind regards,

Hongju He

Academic Editor

PLOS ONE

Journal Requirements:

Reviewers' comments:

Reviewer's Responses to Questions

**Comments to the Author**

1. Is the manuscript technically sound, and do the data support the conclusions?

Reviewer #1: Yes

Reviewer #2: Yes

Reviewer #3: Yes

2. Has the statistical analysis been performed appropriately and rigorously? 

Reviewer #1: Yes

Reviewer #2: Yes

Reviewer #3: Yes

3. Have the authors made all data underlying the findings in their manuscript fully available?

Reviewer #1: Yes

Reviewer #2: Yes

Reviewer #3: Yes

4. Is the manuscript presented in an intelligible fashion and written in standard English?

Reviewer #1: Yes

Reviewer #2: Yes

Reviewer #3: No

5. Review Comments to the Author

**Reviewer #1:** -Re-order key words alphabetically.

-Delete "The primary constituents of cookies include carbohydrates, sugar, and fat, which can contribute to various health issues." in abstract.

-In references, latin names should be italic.

-Some references should be renewed. There are lots of studies about cookies, therefore please cite papers published between 2020-2024.

-Table 7 should be given by a radar graph.

-Please give the ethical permission number for sensory analysis.

-What is the originality of your work? There are lots of enrichment papers available in literature. Why did you choose bitter gourd? Long-term consumption of bitter gourd can have negative health effects. For example, excessive consumption of bitter gourd is associated with miscarriage, liver inflammation, and irregular heartbeat. Also, women who are breastfeeding should also not eat this fruit because some bad substances of bitter melon can be passed into breast milk. People with low blood pressure (or a history of low blood pressure): Bitter melon has the effect of lowering blood pressure thanks to its ingredients Charantin, Polypeptid-P and Vicine.

**Reviewer #2:** The authors investigated the development of oats flour and bitter gourd fortified cookies, specifically on the effects on physicochemical, antioxidant, antimicrobial, and sensory attributes.The findings were very promising, and amply discussed. However, after several readings of it, there are a few areas that would help improve the quality of the work.

a) In the introduction, you talked about flour, oats, individually, without mentioning the process of making cookies using flour. Please, try to squeeze this process in, and condense/shorten the other aspects. Also, support this with a schematic flow diagram that depicts the various stages of cookie making process.

b) Where is actual gap, why is this study relevant, please, strengthen this aspect at the last paragraph of the introduction, very important to

c) The materials and methods is very good, it might be helpful to start this section, that is 2.1, as "Schematic overview of the experimental program", which comprise 4 sentences, first to introduce the flow diagram depicting the key stages of the work, sentence 2 and 3 connecting the objective of this work with the flow diagram, and sentence 4, reiterating that the lab procedures followed appropriate standards, and chemicals used for the analysis were at analytical grade. The reviewer will closely examine this

d)After careful check of results and discussion, the reviewer would encourage authors to do more of discussion, deepen the the why? and how?, and reduce the what?, ok .. Apply your discretion on this, at each section, tell us more about why this happened, and how it is related to existing literature, and if no literature has shown it, how it scientifically happens to be.

Look forward to your revised manuscript

**Reviewer #3:** The manuscript include novelity and has contributed to scientific knwoledge, however there is need to add more crisp in language of the manuscript. It needs to be improved without compromising the scientific intent. There is a need to aff more recent and relevent references. The methodology and dicsussion parts needs to be more comprehsnive to add interest of reader.

I appreciate the efforts of the authors and will be happy to see the changes as suggested

6. PLOS authors have the option to publish the peer review history of their article (what does this mean?). If published, this will include your full peer review and any attached files.

Reviewer #1: No

Reviewer #2: No

Reviewer #3: **Yes: **Dr. Muhammad Farhan Jahangir Chughtai

---

## [Author Response · Author response to Decision Letter 0]

17 Sep 2024

We have carefully addressed all the comments and suggestions provided by the reviewers and the editor. We believe that these revisions have significantly improved the manuscript, and we hope that it will now meet the high standards of PLOS ONE.

---

## [Decision Letter · Decision Letter 1]

13 Dec 2024

Development of oats flour and bitter gourd fortified cookies: effects on physicochemical, antioxidant, antimicrobial, and sensory attributes

PONE-D-24-26098R1

Dear Dr. Zahid,

We’re pleased to inform you that your manuscript has been judged scientifically suitable for publication and will be formally accepted for publication once it meets all outstanding technical requirements.

Kind regards,

Hongju He

Academic Editor

PLOS ONE

Additional Editor Comments (optional):

Reviewers' comments:

Reviewer's Responses to Questions

**Comments to the Author**

1. If the authors have adequately addressed your comments raised in a previous round of review and you feel that this manuscript is now acceptable for publication, you may indicate that here to bypass the “Comments to the Author” section, enter your conflict of interest statement in the “Confidential to Editor” section, and submit your "Accept" recommendation.

Reviewer #2: All comments have been addressed

2. Is the manuscript technically sound, and do the data support the conclusions?

Reviewer #2: Yes

3. Has the statistical analysis been performed appropriately and rigorously? 

Reviewer #2: Yes

4. Have the authors made all data underlying the findings in their manuscript fully available?

Reviewer #2: Yes

5. Is the manuscript presented in an intelligible fashion and written in standard English?

Reviewer #2: Yes

6. Review Comments to the Author

Reviewer #2: After a couple of readings, I am very satisfied with the revised manuscript. It is acceptable for publication.

7. PLOS authors have the option to publish the peer review history of their article (what does this mean?). If published, this will include your full peer review and any attached files.

Reviewer #2: No

---

## [Editor Report · Acceptance letter]

26 Dec 2024

PONE-D-24-26098R1 

PLOS ONE

Dear Dr. Zahid, 

I'm pleased to inform you that your manuscript has been deemed suitable for publication in PLOS ONE. Congratulations! Your manuscript is now being handed over to our production team.

Kind regards, 

on behalf of

Dr. Hongju He 

Academic Editor

PLOS ONE